Interpretable deep learning for the prediction of ICU admission likelihood and mortality of COVID-19 patients

Nazir Amril 1 mohd.nazir@zu.ac.ae
Ampadu Hyacinth Kwadwo 2
1 Department of Information Systems and Technology Management, College of Technological Innovation Zayed University , Abu Dhabi , United Arab Emirates
2 Old Ahinsan, Kumasi-Ghana , Ghana
Wong Ka-Chun
Electronic publication date: 2022 Mar 17
Publication date: 2022
Volume: 8
Electronic Location ID: e889
Received 2021 Aug 16; Accepted 2022 Jan 24
Copyright: © 2022 Nazir and Ampadu
Copyright year: 2022
Copyright holder: Nazir and Ampadu
License: This is an open access article distributed under the terms of the Creative Commons Attribution License, which permits unrestricted use, distribution, reproduction and adaptation in any medium and for any purpose provided that it is properly attributed. For attribution, the original author(s), title, publication source (PeerJ Computer Science) and either DOI or URL of the article must be cited.
License URL: https://creativecommons.org/licenses/by/4.0/

Keywords: Interpretable deep learning, Prediction of ICU admission, Prediction of mortality, COVID-19

Funding: The authors received no funding for this work. The funders had no role in study design, data collection and analysis, decision to publish, or preparation of the manuscript The authors received no funding for this work.

==============================
The global healthcare system is being overburdened by an increasing number of COVID-19 patients. Physicians are having difficulty allocating resources and focusing their attention on high-risk patients, partly due to the difficulty in identifying high-risk patients early. COVID-19 hospitalizations require specialized treatment capabilities and can cause a burden on healthcare resources. Estimating future hospitalization of COVID-19 patients is, therefore, crucial to saving lives. In this paper, an interpretable deep learning model is developed to predict intensive care unit (ICU) admission and mortality of COVID-19 patients. The study comprised of patients from the Stony Brook University Hospital, with patient information such as demographics, comorbidities, symptoms, vital signs, and laboratory tests recorded. The top three predictors of ICU admission were ferritin, diarrhoea, and alamine aminotransferase, and the top predictors for mortality were COPD, ferritin, and myalgia. The proposed model predicted ICU admission with an AUC score of 88.3% and predicted mortality with an AUC score of 96.3%. The proposed model was evaluated against existing model in the literature which achieved an AUC of 72.8% in predicting ICU admission and achieved an AUC of 84.4% in predicting mortality. It can clearly be seen that the model proposed in this paper shows superiority over existing models. The proposed model has the potential to provide tools to frontline doctors to help classify patients in time-bound and resource-limited scenarios.

Introduction

Coronavirus is a virus family that causes respiratory tract illnesses and diseases that can be lethal in some situations, such as SARS and COVID-19. Severe acute respiratory syndrome coronavirus 2 (SARS-CoV-2) is a new type of coronavirus which began spreading in late 2019 in the Chinese province of Hubei, claiming multiple human lives (Li et al., 2020a). The novel coronavirus outbreak was declared a Public Health Emergency of International Concern by the World Health Organization (WHO) in January 2020. The infectious disease caused by the novel coronavirus was given the official title, COVID-19 (Coronavirus Disease 2019) by the WHO in February 2020, and a COVID-19 pandemic was announced in March 2020 by the (World Health Organization; WHO Director General) (Bogoch et al., 2020). Since then, there have been over 170 million cases with many of them being hospitalized. A staggering 3.8 million people died from the disease with the numbers increasing as this paper is being written. Every patient has a different reaction to the virus, with many of them being asymptomatic and a small percentage getting worse rapidly with their organs failing (Leung et al., 2020). The ongoing surge in COVID-19 patients has put a burden on healthcare systems unlike ever before. According to a recent study by Pourhomayoun & Shakibi (2021), once the coronavirus outbreak begins, the healthcare system will be overwhelmed in less than 4 weeks. When a hospitals capacity is exceeded, the death rate rises. The repercussions of an extended stay and increased demand for hospital resources as a result of COVID-19 have been disastrous for health systems around the world, necessitating quick clinical judgments, especially when limited resources are available (Moreira, 2020). COVID-19 infection has been linked to a wide variety of clinical, laboratory, and demographic variables, as described by Rodriguez-Morales et al. (2020) with some of these variables related to an increased risk of critical illness which necessitates admission to an ICU, which may even lead to death. The purpose of this study is to create an interpretable deep-learning algorithm to determine the top predictors of ICU admission and mortality in COVID-19 patients from a vast set of clinical variables collected upon admission. This information will then be used to predict the ICU admission likelihood and mortality of the patients.

Related Work

COVID-19 has been around since late 2019. A lot of studies have gone into different areas to try and find new, useful information about this disease, but just a handful of studies have been done in the hospitalization sector specifically to solve the ICU admission and mortality rate aspect. In this section, we include a detailed account of recent research studies to identify patients more likely to get worse and require ICU care at an early stage. Some well-known prediction approaches can be used for this particular task, and those methods can be divided into two categories: statistical methods, and machine learning methods. Machine learning models have been shown to outperform traditional clinical scoring systems or regression approaches in some situations. Decision tree algorithms or neural networks can detect non-linear relationships between variables which could explain the improved performance. Some of the machine and deep learning approaches used include tree-based algorithms, neural network algorithms, support vector machines, regression algorithms, and the likes.

Manca, Caldiroli & Storti (2020) used a simplified logistic and Gompertz models approach to predict ICU beds and mortality rate for hospital emergency planning in the COVID-19 pandemic for both short term and long term. Their models had two distinct roles to play. They monitored real data and allowed discrimination between models to find the most accurate model, as well as understanding if any unexpected trends emerge. They analyzed their data in two locations, Italy and Lombardy. For the ICU beds dynamic, all the models performed similarly. The Gompertz model performed the best for predicting fatalities in terms of precision and reliability for the whole data, which captures data from the genesis of the pandemic to the 69th day (i.e., April 30, 2020) after the pandemic was announced.

Goic et al. (2021) created a model to forecast ICU beds in Chile in the short term within a 14-day time horizon during times of crisis. They combined autoregressive neural networks, artificial neural networks, and a compartment model to provide a short-term forecast of ICU utilization at the regional level, resulting in the best ICU utilization estimate. Their predictions achieved average forecasting errors of 4%, and 9% for one- and 2-week horizons, respectively, outperforming several other competing forecasting models. Their algorithm captures the epidemiological dynamics of the disease with a compartmental model and is complemented by time-series models that capture short-term changes in the clinical parameters.

Pourhomayoun & Shakibi (2021) used Support Vector Machine (SVM), Artificial Neural Networks (ANN), Random Forest, Decision Tree, Logistic Regression, and K-Nearest Neighbors (KNN) to predict the mortality rate in COVID-19 patients. Their results demonstrated an 89.98% overall accuracy in predicting the mortality rate. The most concerning signs and symptoms for mortality were also determined and described in their work using correlation heat maps. They used a separate data set of COVID-19 patients to assess the accuracy of their proposed model and used a confusion matrix to perform an in-depth analysis of the classifiers and measured the models sensitivity and specificity.

Fernandes et al. (2021) used multipurpose algorithms to predict the likelihood of COVID-19 patients developing critical conditions. Their data set captured data from March to June of the year 2020, consisting of 1,040 patients who had a positive RT-PCR diagnosis for COVID-19 from a large hospital in Sao Paulo, Brazil. From the 1,040 patients, 288 (28%) had a serious prognosis. They trained five machine learning algorithms, namely, artificial neural networks, extra trees, random forests, cat boost, and extreme gradient boosting, using regularly collected laboratory, clinical, and demographic data. They trained the algorithms on a random sample of 70% of patients, leaving 30% for performance evaluation, simulating fresh, unseen data. The algorithms performed extremely well in terms of prediction (average AUROC of 0.92, sensitivity of 0.92, and specificity of 0.82).

Yu et al. (2021) built machine learning algorithms that can predict the need for intensive care and mechanical ventilation. The Random Forest classifier performed the best among the algorithms evaluated, with AUC = 0.80 for predicting ICU needs, and AUC = 0.82 for predicting the need for mechanical ventilation. The data they used consisted of socio-demographic, clinical, and blood panel profiles of patients. They determined the relative importance of blood panel profile data and discovered that when this data was removed from the equation, the AUC decreased by 0.12 units. It provided useful information in predicting the severity of the disease.

Ikemura et al. (2021) aimed to train various machine learning algorithms using automated machine learning (autoML). They chose the model that best estimated how long patients would survive a SARS-CoV-2 infection. They used data that comprised of patients who tested positive for COVID-19 between March 1 and July 3 of the year 2020 with 48 features. The stacked ensemble model (AUPRC = 0.807) was the best model generated using autoML. The gradient boost machine and extreme gradient boost models were the two best independent models, with AUPRCs of 0.803 and 0.793, respectively. The deep learning model (AUPRC = 0.73) performed significantly worse than the other models.

Li et al. (2020b) used the clinical variables of COVID-19 patients to develop a deep learning model and a risk score system to predict ICU admission and mortality of patients in the hospital. The data consisted of 5,766 patients, with comorbidities, vital signs, symptoms, and laboratory tests, between 7th February, 2020 and 4th May, 2020. AUC score was used to evaluate their models. The deep learning model achieved an AUC of 78% for ICU admission and 84% for mortality with the risk score for ICU admission being 72.8% and 84.8% for mortality. Their model was accurate enough to provide doctors with the tools to stratify patients in limited-resource and time-bound scenarios.

Table 1 summarizes the current work being done in this field, the datasets used, the time period carried out in the research and their objectives.

Table 1 Summary of existing works.

Current work	Data sets	Test period	Objective (Covid-19 patients in the hospital)	
Manca, Caldiroli & Storti, 2020	Lombardy, Italy ICU hospital admission	21 February 2020–27 June 2020	Predict ICU beds and mortality rate	
Goic et al. (2021)	Chile official COVID-19 data	May 20th 2020–July 28th 2020	Forecast in the short-term, ICU beds availability	
Pourhomayoun & Shakibi (2021)	Worldwide COVID data from 146 countries	December 1, 2019–February 5th, 2020	Predict the mortality risk in patients	
Fernandes et al. (2021)	São Paulo COVID-19 hospital admission	March 1 2020–28 June 2020	Predict the risk of developing critical conditions	
Yu et al. (2021)	Michigan COVID 19 hospital data	1 February 2020–4 May 2020	Predict the need for mechanical ventilation and mortality.	
Ikemura et al. (2021)	Montefiore Medical Center COVID 19 data	March 1 2020–July 3 2020	Predict patients’ chances of surviving SARS-CoV-2 infection	
Li et al. (2020b)	Stony Brook University Hospital COVID hospital data	7 February 2020–4 May 2020.	Predict ICU admission and in-hospital mortality.	

Existing models used in the literature perform very well for their respective purposes, however, they have a downside in that they are difficult to interpret. The model lacks interpretability on which patient attributes it uses when making a decision (ICU admission and mortality). Existing models use various approaches, but the majority of them use neural network models, which are excellent at achieving good results, but their predictions are not traceable. Tracing a prediction back to which features are significant is difficult, and there is no comprehension of how the output is generated. Therefore, this paper proposes the use of an interpretable neural network approach to predict ICU admission likelihood and mortality rate in COVID-19 patients. It employs a deep learning algorithm that can interpret how the model makes decisions and which features the model selects in making the decision. The model has outstanding and comparable results to other neural network models in the literature. The proposed model can be utilized to generate better outcomes when compared to previously published models.

Method

This paper proposes a high-performance and interpretable deep tabular learning architecture, TabNet, that exploits the benefits of sequential attention (following in a logical order or sequence) to choose features at each decision step which enables interpretability and more efficient learning as the learning capacity is used for the most salient features from the input parameters. The degree to which a human can comprehend the reason for a decision is known as interpretability. The higher the interpretability of the machine or deep learning model, the easier it is for someone to understand why particular decisions or predictions were made. Although neural network models are known to produce excellent results, they have the drawback of being a black box, which means that their predictions are not traceable. It is difficult to trace a prediction back to which features are important, and there is no understanding of how the output was obtained. The interpretabilty here denotes the ability for the model to interpret its decision and shows the features that are the most important in predicting ICU admission and mortality of COVID-19 patients (Ghiringhelli, 2021). This section starts by describing how the input data has been pre-processed for the proposed learning model. Then the different components of the proposed model, and the steps it takes to arrive at a decision to predict ICU admission and mortality has been discussed. Finally, the evaluation metrics used to evaluate the model is analyzed.

Data preprocessing

It is important to pre-process the data before applying it to a machine-learning algorithm. Many pre-processing techniques were applied with each serving a specific purpose. The various pre-processing steps have been discussed below. Various sampling methods were experimented with which included Adaptive Synthetic (ADASYN), and SMOTE to deal with the imbalance in the class labels.

ADASYN (He et al., 2008) is a synthetic data generation algorithm that employs a weighted distribution for distinct minority class examples based on their learning difficulty, with more synthetic data generated for minority class examples that are more difficult to learn compared to minority class examples that are simpler to learn. This is expressed by:

(1) G=(ml−ms)×β

where G in is the total number of synthetic data examples for the minority class that must be produced, ml is the minority class, ms is the majority class, the β is used to determine the desired balance level between 0 and 1.

SMOTE (Chawla et al., 2002) is a technique for balancing class distribution by replicating minority class examples at random:

(2) x′=x+rand(0,1)×|x−xk|

where x′ is the new generated synthetic data, x is the original data, xk is the kth attribute of the data, and rand represents a random number between 0 and 1.

Feature Extraction is a technique for reducing the number of features in a dataset by generating new ones from existing ones. Principal Component Analysis (PCA), Fast Independent Component Analysis (Fast ICA), Factor Analysis, t-Distributed Stochastic Neighbor Embedding t-SNE (t-SNE), and UMAP are the techniques used for the current dataset.

When PCA (Abdi & Williams, 2010) is used, the original data is taken as input and it gives an output of a mix of input features which can better summarize the original data distribution such that its original dimensions are reduced. By looking at pair-wise distances, PCA can maximize variances while minimizing reconstruction error:

(3) cov(X,y)=(1n−1)∑i=1n(Xi−x)(Yi−y)

where x is the input, and y is the output. cov(x, y) is the covariance matrix after which it is transformed to a new subspace which is y = W’x.

FAST ICA (Hyvarinen, 1999) is a linear dimensionality reduction approach that uses the principle of negentropy from maximization of non-gaussian technique as input data and attempts to correctly classify each of them (deleting all the unnecessary noise):

(4) δ=lg(∑i=1N)yi.yiTMSE)

where MSE is the mean squared error, and y is the output.

Factor analysis (Gorsuch, 2013) is a method for compressing a large number of variables into a smaller number of factors. This method takes the highest common variance from all variables and converts it to a single score.

t-SNE (Wattenberg, Viégas & Johnson, 2016) is a non-linear dimensionality reduction algorithm for high-dimensional data exploration. It converts multi-dimensional data into two or more dimensions that can be visualized by humans:

(5) C=KL(P||Q)∑i∑jpijlog(pijqij)

where pij is the joint probability distribution of the features, and qij| is the t-distribution of the features. KL is the Kullback–Leiber divergence, P and Q are the distribution in space.

UMAP (McInnes, Healy & Melville, 2018) is a general-purpose dimensionality reduction technique that can be used to pre-process the input data for machine learning. The theoretical foundation for UMAP is focused on Riemannian geometry and algebraic topology:

(6) pi|j=e−d(xi,xj)−piσi

where p represents the distance from each ith data point to the nearest jth data point.

Before feeding all this information to the TabNet to act on, the dataset needs to be split into a training set and a testing set. The training dataset is used to train the model and the testing dataset is used to evaluate the models performance. The Stratified K-fold (Zeng & Martinez, 2000) cross-validation was implemented which splits the data into “k” portions. In each of “k” iterations, one portion is used as the test set, while the remaining portions are used for training. The fold used here was k = 5 which means that the dataset was divided into 5 folds with each fold being utilized once as a testing set, with the remaining k − 1 folds becoming the training set. This ensures that no value in the training and test sets is over- or under-represented, resulting in a more accurate estimate of performance/error.

Architecture of TabNet model

A TabNet Model (Arik & Pfister, 2019) is proposed to perform prediction of ICU admission and mortality parameters. A schematic diagram of the proposed TabNet deep learning model is presented in Fig. 1.

Figure 1 TabNet model architecture.

The TabNet model consists primarily of sequential multi-steps that transfer inputs from one stage to the next. There are three key layers of this model, namely, the Feature Transformer, the Attentive Transformer, and the Mask. The Feature Transformer is made up of four sequential Gated Linear Unit (GLU) decision blocks, which allow the selection of important features for predicting the next decision. The Attentive Transformer provides sparse feature selection that uses sparse-matrix to improve interpretability and learning. The way it does this is by giving importance to the most important features. The mask is then used in conjunction with the Transformer to produce two decision parameters n(d) and n(a) which are then passed on to the next step. n(d) is the output decision that predicts the two classes, namely, ICU admission (yes/no) and Deaths (yes/no). n(a) is the input to the next Attentive Transformer, where the next cycle starts. From the Feature transformer, the output is sent to the Split module.

From Fig. 1, the input features are Batch Normalized (BN) and passed to the Feature Transformer, where it goes through four layers of a Fully Connected layer (FC), a Batch Normalization layer (BN), and a Gated Linear Unit (GLU) in that order. The Feature Transformer produces n(d) and n(a). The Feature Transformer comprises two decision steps.

A Fully Connected (FC) layer is a type of layer where every neuron is connected to every other neuron. The output from the FC layer should always be Batch Normalized. Batch normalization is used to transform the input features to a common scale. It can be represented mathematically as:

(7) BN=x−μbω2+ϵ

where x represents the input features, μb denotes the mean of the features and ω2 denotes the variance.

The fully connected layer is the combination of all the inputs with the weights, which can be represented mathematically as:

(8) FC=W(x)

where x denotes the input features, and W denotes the weights.

These operations are done sequentially, starting from Eq. (9), then to Eq. (10) and finally to Eq. (11)

(9) FC=W(x)

(10) BN=x−μbω2+ϵ

The Gated Linear Unit (GLU) is simply the sigmoid of x:

(11) GLU=σ(x)

The Fully Connected layer performs its operations, then its output is fed into the Batch Normalization layer to perform its operations. Finally, the output from the Batch Normalization is fed into the Gated Linear Unit, all in a sequential manner.

The decision output from the Feature Transformers n(d) is also aggregated and embedded in this form, and a linear mapping is applied to get the final decision. As a result, they are made up of two shared decision steps and two independent decision steps. A residual connection connects the shared steps with the independent steps and they are summed together via the ⊕ operation, which is a direct summation block.

Since the same input features in the dataset are used in distinct steps, the layers are shared between two decision steps for robust learning. By ensuring that the variation across the network does not change significantly, normalization with a square root of 0.5 helps to stabilize learning which produces the outputs of n(d) and n(a) as mentioned previously.

From the Feature Transformer, and after the Split layer, the Attentive Transformer is applied to determine the various features and their values. The feature importance for that step is combined with the other steps and is made up of four layers: FC, BN, Prior Scales, and Sparse Max in sequential order. The split layer splits the output and obtains p[i − 1], which is then passed through the Fully Connected (FC) layer and the Batch Normalization (BN) layer, whose purpose is to achieve a linear combination of features allowing higher-dimensional and abstract features to be extracted. The output from the BN layer is multiplied using the tensor product ⊗ with the previous decision steps Prior Scale p[i − 1]. The Prior Scale represents the employment of features in previous decision steps. If gamma (γ) is set to 1, all features have the same importance in predicting ICU admission and mortality. Sparse Max is then used to generate M[i]. The process of learning a Mask (Martins & Astudillo, 2016) is then represented by:

(12) M[i]=Sparsemax(P[i−1]∗hi(p[i−1])

where the hi represents the summation of the FC and the BN layer, the p[i − 1] represents the division by the split layer in the previous step, and P[i − 1] represents the Prior Scales. By transferring the Euclidean projection onto the probabilistic simplex, Sparse Max encourages sparsity and makes feature selections sparser. Sparse Max implements weight distribution for each feature of each sample and sets the sum of the weights of all features of each sample to 1 (Yoon, Jordon & van der Schaar, 2018), allowing TabNet to employ the most useful features for the model in each decision step. M[i] then updates p[i]:

(13) P[i]=∏j=1i(γ−Mj)

If γ is set closer to 1, the model uses different features at each step; if γ is greater than 1, the model uses the same features in multiple steps. The Sparse matrix is similar to the softmax, except that instead of all features adding up to 1, some will be 0 and others will add up to 1. The Sparse Max is expressed as:

(14) ∫i=1nsparsemax(x)i

This makes it possible to choose features on an instance-by-instance basis with various features being considered at various steps. These are then fed into the Mask layer, which aids in the identification of the desired features. The Feature Transformer is applied again, and the resulting output is split to the Attentive Transformer. The split layer divides the output from the feature transformer into two parts which are d[i], and a[i]:

(15) d[i],a[i]=fi(M[i]∗f)

where d[i] is used to calculate the final output of the model, and a[i] is used to determine the mask of the next step. ReLU activation is then applied:

(16) f(x)=max(0,d[i])

where f(x) returns 0 if it receives any negative input, but for any positive value x, it returns that value back. The contribution of the ith step to obtain the final result can be expressed as:

(17) ϕb[i]=∫c=1NdRelu(d[i],c[i])

where ϕb[i] indicates the features that are selected at the ith step.

To map the output dimension, the outputs of all decision steps are summed and passed through a Fully Connected layer. Combining the Masks at various stages necessitates the use of a coefficient that can account for the relative value of each step in the decision-making process. The importance of the features can be expressed using the equation:

(18) Magg−b,j=∫i=1Nstepsϕb[i]Mb,j∫j=1D∫i=1Nstepsϕb[i]Mb,j

TabNet decision making process

Figure 2 provides an illustration of how the TabNet (Arik & Pfister, 2019) makes a decision (individual explainability). TabNet has a feature value output called Masks, which shows whether a feature is selected at a given decision step in the model and can be used to calculate the feature importance. The masks for each input feature are represented by each row, and the column represents a sample from the data set. Brighter colors show a higher value. Consider Fig. 2, where nine features ranging from feat 0 to feat 8 are shown. For the random sample at 3, the first feature is the one being heavily used, hence the brighter colour, and the sample at 6, three features have brighter colours, feature 0, 1 and 8, with 8 being the brightest, signifying the feature 8’s output was heavily used for this sample.

Figure 2 TabNet decision making process.

The mask value for a given sample indicates how significant the corresponding feature is for that sample. Brighter columns indicate the features that contribute a lot to the decision-making process. It can be seen that the majority values for features other than 0, 1, 4, 5, and 8 are close to ‘0,’ indicating that the TabNet model correctly selects the salient features for the output. We can then interpret which features the model selects enhancing the interpretability of the model. With this, the features that contribute to individuals being admitted to the ICU and dying of the COVID-19 disease can be ranked accordingly.

Evaluation metrics

Evaluation metrics are the metrics used to evaluate the model to determine if it is working as well as it should be. The evaluation metrics used are:

Confusion matrix describes a classification models output on a collection of test data for which the true values are known. The matrix compares the actual target values to the machine learning models predictions. It includes true positives (TP) which are correctly predicted positive values, indicating that the value of the real class and the value of the predicted class are both yes, True negatives (TN) which correctly estimates negative values, indicating that the real class value is zero and the predicted class value is zero as well. False positives (FP), where the actual class is no but the predicted class is yes. False negatives (FN), where the actual class is yes but the predicted class is no.

Accuracy is the proportion of correctly expected observations to all observations.

(19) Accuracy=TP+TNTP+FP+FN+TN

Precision is the ratio of correctly predicted positive observations to total predicted positive observations.

(20) Precision=TPTP+FP

Recall is the ratio of correctly expected positive observations to all observations in the actual class.

(21) Recall=TPTP+FN

F1 Score is the weighted average of Precision and Recall.

(22) F1Score=2×(Precision×Recall)Recall+Precision

The various mathematical notations used in this section are shown in Table 2.

Table 2 Notations.

Notations	Definitions	
x	Features	
σ	sigmoid	
n(d)	output decision from current step	
n(a)	input decision to the next current step	
W	weights	
⊕	direct summation	
otimes	tensor product	
γ	gamma	
β	beta	
μ b	mean	
∫i=1i	integral block	
∏i=1i	product block	
P[i−1]	prior scales	
p[i−1]	split layer division	
h i	FC layer + BN layer	
M j	Mask learning process	
d[i]	final output	
a[i]	determine mask of next step	
f(x)	function to return value of relu function	
ϕ	features selected at ith step	
Maggb,−j	importance of features	
G	Total number of synthetic data examples	
m l	minority class	
m s	majority class	
β	Desired balance level	
x′	new generated synthetic data	
cov(X, y)	covariance matrix	
pi|j	joint probability distribution of features	
qi j	t-distribution of features	
KL	Kullback–Leiber divergence	
TP	True positive	
TN	True negative	
FP	False positive	
FN	False negative	

Experimental Results

To demonstrate the effectiveness of the method in predicting ICU admission and mortality of patients, the different TabNet model hyperparameters, dimensionality reduction techniques, and oversampling methods have been thoroughly examined and contrasted. The section starts with the description of datasets followed by the statistical analysis, and it concludes with results and discussion.

Description of data sets

There are two data sets used for this analysis, one for the ICU likelihood and the other for the mortality rate. The ICU data set consists of 1,020 individuals with 43 features and the mortality data set consists of 1,106 individuals with 43 features. The features consist of vital signs, laboratory tests, symptoms, and demographics of these individuals. There are two labels associated with the ICU dataset which are, ICU admitted (label 0), and ICU non-admitted (label 1). Similarly, there are two labels associated with the death dataset which are, non-death (label 0), and death (label 1). The datasets are unbalanced with distribution ratios of 75.5:24.5, and 86.1:13.9 for the ICU and mortality datasets, respectively. Table 3 shows the summary of the description of the datasets.

Table 3 Description of datasets.

Dataset	No. patients-No. features	Class labels	Class distribution ratio (Pos: Neg)	
ICUMice-ICU	1,106-43	1 = death 0 = non-death	86.1:13.9	
DEADMice-Mortality	1,020-43	1 = ICU 0 = no-ICU	75.5:24.5	

Statistical analysis

ICU admission

There were more males than females in the study population. The non-Hispanic ethnicity forms the most individuals, and the Caucasian race has the highest number of individuals in the population. Hypertension is the co-morbidity that most individuals in the study population presented, with cancer being the least. The average age of individuals that needed the ICU is higher than those that did not. Regarding the vital signs, heart rate is the sign that showed quite a big difference on average, with the individuals needing ICU having a higher heart rate than their fellow counterparts. Procalcitonin, Ferritin and C-reactive protein are the laboratory findings that showed the biggest average difference, with individuals needing ICU showing higher levels of these. Table 4 shows the demographics, vital signs, comorbidities, and laboratory discoveries of ICU patients and non-ICU patients.

Table 4 Relationship between features and ICU admission.

Features/variables	ICU (n = 271)	No-ICU (n = 835)	
	Demographics		
Age, mean	59.42	62.06	
Male	67.5% (183)	54% (451)	
Female	32.5% (88)	46% (384)	
	Ethnicity		
Hispanic/Latino	28.8% (78)	26.6% (222)	
Non-Hispanic/Latino	54.6% (148)	60.7% (507)	
Unknown 16.6% (45)	12.7% (106)		
	Race		
Caucasian	45.4% (123)	54.3% (453)	
African American	4.79% (13)	7.3% (61)	
American Indian	0.7% (2)	0.2% (2)	
Asian	7.4% (20)	3.1% (26)	
Native Hawaiian	0	0.1% (1)	
More than one race	0	0.6% (5)	
Unknown/not reported	41.7% (113)	34.4% (287)	
	Comorbidities		
Smoking history	22.5% (61)	25.6% (214)	
Diabetes	29.5% (80)	26.3% (220)	
Hypertension	46.5% (126)	49.3% (412)	
Asthma	8.5% (23)	5.1% (43)	
COPD	6.3% (17)	9.1% (76)	
Coronary artery disease	14.4% (39)	15.1% (126)	
Heart failure	6.6% (18)	7.4% (62)	
Cancer	5.5% (15)	10.5% (88)	
Chronic kidney disease	7.4% (20)	9.7% (81)	
	Vital signs		
Systolic blood pressure (mmHg), mean	124.8	128.99	
Temperature (degree Celsius), mean	37.63	37.47	
Heart rate, mean	106.1	98.2	
Respiratory rate (rate/min), mean	25.28	21.77	
	Laboratory Findings		
Alanine aminotransferase (U/L), mean	49.62	47.03	
C-reactive protein (mg/dL), mean	15.4	9.49	
D-dimer (ng/mL), mean	1,101.92	1,210.51	
Ferritin (ng/mL), mean	1,469.67	1,005.43	
Lactase dehydrogenase (U/L), mean	481.7	377.85	
Lymphocytes (*1,000/ml)	12.43	14.85	
Procalcitonin (ng/mL), mean	2.66	0.97	
Troponin (ng/mL), mean	0.038	0.03	

For the symptoms, loss of smell and loss of taste were the symptoms that most individuals that got admitted to the ICU acquired whereas fever and cough were the symptoms that the least number of individuals acquired to be admitted to the ICU. Overall, over 70% of individuals acquired a symptom of disease at admission to the ICU. Table 5 shows the relationship between symptoms and ICU admission by looking at the distribution of patients who were admitted to the ICU and whether or not they had a symptom.

Table 5 Relationship between symptoms and ICU admission.

Symptoms	Percentage of patients with symptoms (%)	
Fever	70.5	
Cough	70.5	
Shortness of Breath (SOB)	77.5	
Fatigue	79.3	
Sputum	90.77	
Myalgia	77.5	
Diarrhea	77.9	
Nausea or vomiting	83.3	
Sore throat	92.3	
Runny nose or Nasal congestion	94.8	
Loss of smell	95.9	
Loss of Taste	95.6	
Headache	89.7	
Chest discomfort or chest pain	84.1	

Certain symptoms had a higher correlation with ICU admission than others and Table 6 gives a summary of this. It is observed that the Shortness of Breath (SOB) feature has the highest correlation with admission to the ICU unit.

Table 6 Correlation between symptoms and ICU admission.

Symptoms	Correlation (Pearson)	P values	
Fever	0.046	0.122	
Cough	0.028	0.348	
Shortness of Breath (SOB)	0.1	0.0008	
Fatigue	−0.03	0.248	
Sputum	0.055	0.065	
Myalgia	−0.005	0.869	
Diarrhea	−0.018	0.5415	
Nausea or vomiting	−0.035	0.247	
Sore throat	0.009	0.757	
Runny nose or Nasal congestion	0.0177	0.556	
Loss of smell	−0.0003	0.992	
Loss of Taste	−0.012	0.689	
Headache	0.013	0.673	
Chest discomfort or chest pain	−0.0007	0.98	

Mortality

There were more males than females in the study population. The non-Hispanic ethnicity forms the most individuals and the Caucasian race has the highest number of individuals in the population. Hypertension is the co-morbidity that most individuals in the study population presented, with Asthma being the least. The average age of individuals that died is higher than those that did not. Regarding the vital signs, Respiratory rate is the sign that showed quite a big difference on average, with the individuals that died having a higher respiratory rate compared to the ones who did not die. Procalcitonin, Ferritin, D-dimer, and C-reactive protein are the laboratory findings that showed the biggest average differences, with individuals that died showing higher levels of these. Table 7 shows the demographics, vital signs, comorbidities, and laboratory discoveries of patients that died and the ones who did not die.

Table 7 Relationship between features and mortality.

Features/variables	Death (n = 271)	No-Death (n = 835)	
	Demographics		
Age, mean	73	59.83	
Male	65.5% (93)	55% (483)	
Female	34.5% (49)	45% (395)	
	Ethnicity		
Hispanic/Latino	16.2% (23)	28.5% (250)	
Non-Hispanic/Latino	73.9% (105)	57.4% (504)	
Unknown	9.9% (14)	14.1% (124)	
	Race		
Caucasian	64.1% (91)	51.3% (450)	
African American	4.2% (6)	6.9% (61)	
Asian	6.3% (9)	3.% (33)	
American Indian	0.7% (2)	0.23% (2)	
Native Hawaiian	0	0.1% (1)	
More than one race	0	0.6% (5)	
Unknown/not reported	24.6% (35)	37.1% (326)	
	Comorbidities		
Smoking history	36.6% (52)	23.2% (204)	
Diabetes	33.8% (48)	26.08% (229)	
Hypertension	64.8% (92)	45.8% (402)	
Asthma	4.22% (6)	5.8% (51)	
COPD	16.2% (23)	7.5% (66)	
Coronary artery disease	27.5% (39)	13.1% (115)	
Heart failure	20.4% (29)	5.4% (47)	
Cancer	13.4% (19)	8.9% (78)	
Immunosuppression	5.6% (8)	7.4% (65)	
Chronic kidney disease	14.08% (20)	8.5% (75)	
	Vital signs		
Systolic blood pressure (mmHg), mean	127.45	128.57	
Temperature (degree Celsius), mean	37.3	37.52	
Heart rate, mean	98.28	100.38	
Respiratory rate (rate/min), mean	26.39	21.79	
	Laboratory Findings		
Alanine aminotransferase (U/L), mean	42.91	48.45	
C-reactive protein (mg/dL), mean	16.07	9.62	
D-dimer (ng/mL), mean	2,626.27	1,016	
Ferritin (ng/mL), mean	1,565	1,037.5	
Lactase dehydrogenase (U/L), mean	588.28	363.14	
Lymphocytes (*1,000/ml)	10.96	14.99	
Procalcitonin (ng/mL), mean	5.14	0.76	
Troponin (ng/mL), mean	0.07	0.0278	

Loss of smell and loss of taste were the symptoms that most individuals that died acquired, whereas fever and cough were the symptoms that the least number of individuals that died acquired. Overall, at least 50% of individuals acquired a particular symptom before they died. Table 8 shows the relationship between symptoms and mortality by looking at the distribution of patients that died and whether or not they had a symptom.

Table 8 Relationship between symptoms and mortality.

Symptoms	Percentage of patients with symptoms (%)	
Fever	57	
Cough	51.4	
Shortness of Breath (SOB)	71.8	
Fatigue	86.6	
Sputum	93	
Myalgia	89.44	
Diarrhea	81	
Nausea or vomiting	93	
Sore throat	95.1	
Runny nose or Nasal congestion	97.18	
Loss of smell	98.59	
Loss of Taste	98.59	
Headache	95.07	
Chest discomfort or chest pain	92.96	

Certain symptoms had a higher correlation with mortality than others and Table 9 gives a summary of this. It is observed that the headache feature has the highest correlation with death.

Table 9 Correlation between symptoms and mortality.

Symptoms	Correlation (Pearson)	P values	
Fever	−0.08	0.009	
Cough	−0.149	0.0000069	
Shortness of Breath (SOB)	0.031	0.32	
Fatigue	−0.09	0.003	
Sputum	0.006	0.84	
Myalgia	−0.119	0.00013	
Diarrhea	−0.04	0.199	
Nausea or vomiting	−0.128	0.00004116	
Sore throat	−0.037	0.233	
Runny nose or Nasal congestion	−0.03	0.318	
Loss of smell	−0.05	0.0965	
Loss of Taste	−0.066	0.0377	
Headache	0.062	0.048	
Chest discomfort or chest pain	−0.1	0.002	

Experimental settings

Hyperparameters of TabNet

The TabNet model has a considerable number of hyperparameters which can be tuned to improve performance. The TabNet comes with some default parameters which works well, but for certain use cases, different values of certain hyperparameters yield better performances. Table 10 shows the default hyperparameters of the TabNet.

Table 10 Default hyperparameters of the TabNet Model.

Training hyper parameters	Default values	
Max epochs	200	
Batch Size	1,024	
Masking Function	sparsemax	
Width of decision prediction layer	8	
Patience	15	
momentum	0.02	
n shared	2	
n independent	2	
gamma	1.3	
nsteps	3	
lambda sparse	1e−3	

Max epochs are the maximum number of epochs for training. It can be any number equal to or higher than 10. Batch size is the number of examples per batch. The number should preferably be a multiple of two and be greater than 16. The masking function is used for selecting features. Higher values for the width of decision prediction layer gives more capacity to the model with the risk of overfitting. The values typically range from 8 to 64. Patience is the number of consecutive epochs without improvement before performing an early stoppage. If patience is set to 0, then no early stopping will be performed. Momentum for batch normalization typically ranges from 0.01 to 0.4. n shared is the number of shared Gated Linear Units at each step. The usual values range from 1 to 5. n independent is the number of independent Gated Linear Units layers at each step. The usual values range from 1 to 5. Gamma is the coefficient of feature re-usage in the masks. Its values range from 1.0 to 2.0. A value close to 1 will make mask selection less correlated between layers. n steps is the number of steps in the architecture (usually between 3 and 10). lambda sparse is the extra sparsity loss coefficient. The bigger the coefficient, the sparser the model will be in terms of feature selection (Arik & Pfister, 2019).

Results and analysis

We present a summary of our experimental results and analysis on two categories: ICU Admission and Mortality.

ICU admission

Results of Hyperparameter tuning using TabNet

The hyper parameters of the model have been tuned using various values of each parameter. The final table which has the best hyper parameters in predicting ICU admission for the various metrics are shown here, the rest of the tables for the individual hyper parameters can be seen in the appendix section. In varying the width of decision prediction layer (nd), the value of nd was changed from a range of 2 to 64 to determine the best output. The results were the best when nd was set to 64.

In varying number of steps in the architecture (nsteps), the value of nsteps was varied from 3 to 12 to determine the best output. A value of 3 gave the best results. Changing the nsteps to numbers between 8 and 12 showed a slight decrease in performance which indicates that the performance will not be enhanced by increasing the number of steps.

In varying gamma. Changing the gamma shows a very haphazard trend in performance, the best results are given when the gamma is 2.0. Increasing the gamma does not improve the results. Thus, gamma was not increased any further.

In varying number of independent gates (nindependent), the number of independent gates was varied from two to seven. All the nindependent gates obtained similar results converging to the best result with 2. Thus, two independent gates give the best results.

In varying number of shared gates (nshared), the nshared gates was varied from two to seven, with two gates achieving the best results. Increasing the gates did not improve the results.

In varying values of momentum, momentum values of 0.2 and 0.3 displayed the best results, with higher values of momentum producing poorer results.

In varying lambda sparse, the values of lambda sparse was varied from 0.001 to 0.005. The results of the model showed a negative correlation with the value of lambda sparse. The best result was achieved with a value of 0.001.

Two different types of masks were used, the entmax and the sparsemax. It was concluded that the mask type of entmax gives a better result across the board in all the performance metrics.

The number of epochs and stopping condition was also experimented to determine the impact it has on the performance of the TabNet model. The results are generally better when the stopping condition is defined. The best results are achieved with an epoch of 150, and patience greater than 60. The results do not change when the patience is greater than 60.

Regarding the dimensionality reduction methods, five methods were experimented with to check its effect on the performance of the TabNet baseline model. The results using the Fast ICA has the best results with it falling short only on recall to PCA. The most important parameter in the table is the AUC, in which the Fast ICA has a score of 83.6.

Both PCA method and the Fast ICA methods yielded similar scores on the impact of different dimensionality reduction methods on the performance of the best TabNet model. For the AUC parameter, the Fast ICA has the highest score of 86.4.

Different oversampling methods on the performance of the TabNet baseline model were experimented with and the results using the ADASYN method has the best outcome in all the measured performance metrics.

The impact of different oversampling methods on the performance of the TabNet Best model was determined. The results using the ADASYN method has the best results in all the measured performance metric.

The tables showing the various experimentations of the individual hyperparameter explained above can be found in the appendix section.

Table 11 shows the performance of the TabNet Baseline, and TabNet Best models with FastICA dimensionality reduction, and ADASYN oversampling method. The results of the TabNet Best is the best amongst the other baseline models.

Table 11 Performance of the most optimized TabNet model with corresponding standard deviations across all the runs.

Model	AUC	F1Score	Accuracy	Recall	Precision	
Li et al. (2020b) (baseline)	72.8	55.1	72.1	76.0	43.2	
TabNet Baseline+Fast ICA+ADASYN	79.77 ± 1.87	80.09 ± 2.78	82.1 ± 2.05	84.47 ± 6.57	77.01 ± 4.71	
TabNet Best+ Fast ICA+ ADASYN	84.66 ± 2.46	85.73 ± 3.208	84.52 ± 3.07	92.31 ± 1.08	81.28 ± 4.87	

Figure 3 shows the trend of the various hyperparameter during the experiments and tuning. Figure 4 further shows the feature importance masks for predicting ICU admission. TabNet features a feature value output called Masks that may be used to quantify feature importance and indicate if a feature is chosen at a particular decision step in the model. Each row represents the masks for each input element and the column represents a sample from the dataset. The brighter the color, the higher the value. In predicting ICU admission, two of the masks are shown as an example in Fig. 4, where the features which the respective masks are paying attention to can be seen in bright colors. The brighter a grid, the more important that particular feature is for the particular Mask. The number of grids lighting up corresponds to the number of features that are being paid attention to by the particular Mask. It can be seen that Mask 0 is paying the most attention to the earlier features, with an emphasis on the 20th feature. Mask 1 is paying the most attention to the later features, with the most attention given to features 32 and 38. The average feature output among all the Masks is used to arrive at the final decision.

Figure 3 Varying hyper parameters with respect to AUC score for predicting ICU admission.

Figure 4 Feature importance masks for predicting ICU admission using TabNet (Individual interpretability).

Figure 5 shows a graph of features against feature importance. Features are the symptoms that contribute to an individual being admitted to the ICU. Feature importance stands for the importance of the symptom in contributing to the patient being admitted into the ICU, where a larger number indicates a higher contribution to an ICU admission. All features have some importance in determining if a person would be admitted to the ICU. The sum of all the feature importance data points is 1. The top 5 features that contribute greatly to a person needing to be admitted to the ICU were Ferritin, ALT, ckdhx, Diarrhoea and carcinomahx.

Figure 5 Feature importance for predicting ICU admission using TabNet (Global interpretability).

Best final model for ICU prediction

The model was analyzed and its output compared with different TabNet configurations based on different feature extractors. The model with the best results has been selected as the final proposed model for ICU prediction. The proposed model is a TabNet model with 150 epochs, 128 batch size, and 60 patience with the number of steps of 2, width of precision of layer of 64, gamma of 1.3, entmax mask type, n independent of 2, momentum of 0.3, lambda sparse of 1e–3, and n shared of 2, using the Fast Independent Component Analysis as the feature extractor, and ADASYN as the sampling technique to balance the imbalanced data. Figures 6 and 7 below show the loss graph, and training and validating accuracy graph of the TabNet in predicting ICU admission.

Figure 6 Model Loss for best TabNet model.

Figure 7 Training and validation accuracy for best TabNet model.

Figure 6 shows a graph of loss against number of epochs. The losses reduce as the number of epochs increases. At the first epoch, the model has not learned from the data, so the margin of error is big. As the model starts to learn (goes through the epochs), the error reduces and hence the loss reduces.

Figure 7 shows a graph of accuracy against number of epochs. Accuracy tends to go higher as the number of epochs increases. At the early stages of the training, the accuracy is low, but as the model begins to learn the patterns of the data, the accuracy increases and reaches a higher value at the end of the epochs (150). Difference between the training accuracy and the testing accuracy is not high, which suggests that the model is not overfitting on the dataset. The precision-recall curve, which demonstrates a trade-off between the recall score (True Positive Rate), and the precision score (Positive Predictive Value), is used for this analysis due to the dataset being imbalanced (there is a large skew in the class distribution). The confusion matrix also gives a sense of the specific number of patients that were correctly classified as needing ICU or not needing it, and the ones that were incorrectly classified.

A large area under the curve indicates high recall and precision, with high precision indicating a low false-positive rate and high recall indicating a low false-negative rate. High scores for both indicate that the classifier is producing correct (high precision) results as well as a majority of all positive outcomes (high recall). From Figure 8, it can be seen that there is a large area under the curve, indicating that the model is functioning very well. With an AUC of 88.75%, the model can distinguish between most of the ICU and the No-ICU patients. It can be seen from the confusion matrix in Fig. 9 that the model correctly predicted 61 individuals needing the ICU and 84 individuals not needing the ICU. A total of 18 individuals were incorrectly classified as not needing the ICU when they needed the ICU and 5 individuals as needing the ICU when they did not need the ICU.

Figure 8 Precision-Recall curve of the best TabNet model for predicting ICU admission.

Figure 9 Confusion matrix of the best TabNet model for predicting ICU admission.

With the proposed model doing an excellent job in predicting ICU admissions, the proposed model is compared to the models existing in the literature.

It can be seen from Table 12 that the proposed model beats the model reported by Li et al. (2020b), in all metrics which is a clear suggestion that the proposed model is superior.

Table 12 Comparison of results between proposed method and existing technique.

Model	AUC	F1Score	Accuracy	Recall	Precision	
Li et al. (2020b) (baseline)	72.8	55.1	72.1	76.0	43.2	
Proposed method	88.4	89.7	88.7	93.3	86.4	

Mortality

We now present a summary of our experimental results and analysis for the Mortality category.

Results of Hyperparameter tuning using TabNet

Similarly in predicting morality, the hyper parameters of the model have been tuned using various values of each parameter. Figure 10 shows a graph of how varying the various hyper parameter values affect the mortality. In varying the width of decision prediction layer (nd), the value of nd was changed from a range of 2 to 64 to determine the best output. The results were the best when nd was set to 8.

Figure 10 Varying Hyper parameters with respect to AUC score for predicting mortality.

In varying number of steps in the architecture (nsteps), the value of nsteps was varied from 3 to 12 to determine the best output. A value of 3 gave the best results. nsteps of 10 to 12 did not show any improvement in results, which indicates that the performance of the model will not be improved by increasing the number of steps.

In varying gamma, the values of gamma was varied from 1.3 to 2.2. The performance of the model shows a very haphazard trend with the changing values. A gamma of 2.0 gives the best results.

In varying number of independent gates (nindependent), the number of independent gates was varied from 2 to 7. The number of gates which gave the best output is 2, and increasing the number of gates decreased the accuracy of the results.

In varying number of shared gates (nshared), the number of shared gates was varied from 2 to 7. The number of shared gates of 2 gave the best results and increasing the number of gates did not improve the results. Although there is a spike in results when the number of shared gates is 5, the performance reduces when it is increased further.

The values of momentum was varied from 0.02 to 0.3, with 0.02 giving the best results. Increasing the value of the momentum gave poorer results.

In varying lambda sparse, the values of lambda sparse was varied from 0.001 to 0.005, with 0.001 achieving the best results. The value of the lambda sparse had a negative correlation with the performance of the model.

The different masktypes used were sparsemax, and entmax. The output using the sparsemax had a better result compared to the entmax.

The number of epochs and stopping condition was also experimented to determine the impact it has on the performance of the TabNet model. The results are generally better when the stopping condition is defined. The best results are achieved with an epoch of 150, and patience greater than 60. The results do not change when the patience is greater than 60.

Regarding the dimensionality reduction methods, five methods were experimented with to check its effect on the performance of the TabNet Baseline model. The results using the Fast ICA has the best results with it falling short only on recall to PCA. The most important parameter in the table is the AUC, in which the PCA has a score of 94.0.

The Fast ICA yielded the best score on the impact of different dimensionality reduction methods on the performance of the best TabNet model. For the AUC parameter, the Fast ICA has the highest score of 86.4.

Different oversampling methods on the performance of the TabNet Baseline model were experimented with and the results using the ADASYN method has the best outcome in all the measured performance metrics.

The impact of different oversampling methods on the performance of the TabNet Best model was determined. The results using the ADASYN method has the best results in all the measured performance metric.

The tables showing the various experimentations of the individual hyperparameter explained above can be found in the appendix section.

Table 13 shows the performance of the TabNet Baseline, and TabNet Best models with FastICA dimensionality reduction, and ADASYN oversampling method. The results of the TabNet Best is the best amongst the other baseline models.

Table 13 Performance of the best final TabNet model with FastICA dimensionality reduction method and ADASYN oversampling method.

Model	AUC	F1Score	Accuracy	Recall	Precision	
Li et al. (2020b) (baseline)	84.4	61.6	85.3	70.6	52.2	
TabNet Baseline+Fast ICA+ADASYN	89.03 ± 2.19	89.12 ± 2.40	88.92 ± 2.40	92.98 ± 2.97	85.75 ± 4.11	
TabNet Best+Fast ICA+ADASYN	91.59 ± 1.63	91.74 ± 1.63	91.49 ± 1.62	96.65 ± 1.91	87.36 ± 2.24	

Figure 11 also shows how the TabNet model makes decisions to predict mortality. In predicting mortality, two of the Masks are shown as an example in Fig. 11. The color of the grid determines the importance of the particular feature to the particular Mask. Brightness of the colors correspond to the importance of the feature. It can be seen that Mask 0 is paying attention to a couple of features but most of its attention is on the last feature, and Mask 1 is paying attention to only 4 features, but the most attention is on the 6th and 15th. Again, the average feature output among all the Masks is used to arrive at the final decision.

Figure 11 Feature importance masks for predicting mortality (Individual interpretability).

Figure 12 shows a graph of features against feature importance. The features are the symptoms that contribute to an individual dying of COVID-19. Feature importance stands for the importance of the feature to contribute to the death of a person, where a larger number corresponds to a higher contribution to death. All features have some importance in determining if COVID-19 would be fatal to a person. The feature importance of all the features add up to 1. The top 5 features which contributes greatly to a person dying of COVID-19 were COPD, Ferritin, Myalgia, coronary artery diseases, and CRP.

Figure 12 Feature importance for predicting mortality (Global Interpretability).

Best final model for mortality prediction

The model was analyzed and its outputs compared with different TabNet configurations based on different feature extractors. The model with the best results was selected. The final model is the TabNet model with 150 epochs, 128 batch size and 60 patience with the number of steps of 3, width of precision of layer of 8, gamma of 1.7, sparsemax mask type, n independent of 2, the momentum of 0.02, lambda sparse of 1e−3 and n shared of 2, using the Fast Independent Component Analysis as the feature extractor and ADASYN as the sampling technique to balance the imbalanced data. Figures 13 and 14 below show the loss graph, and training and validating accuracy graph of the TabNet in predicting Mortality.

Figure 13 Model Loss for best TabNet model.

Figure 14 Training and validation accuracy for best TabNet model.

Figure 13 shows a graph of loss against number of epochs. Losses reduce as the number of epochs increases. At the first epoch, the model has not learned a lot from the data, so the margin of error is big. As the model is learning (going through the epochs), the error reduces and hence the loss reduces.

Figure 14 shows a graph of accuracy against number of epochs. The accuracy tends to increase as the number of epochs increases. At the early stages of the training, the accuracy is low, but as the model begins to learn the patterns of the data, the accuracy increases and reaches a higher value at the end of the epochs (150). The difference between the training accuracy and testing accuracy is not high which suggests that the model does not overfit on the dataset.

In predicting mortality also, the ROC curve which demonstrates a trade-off between the true positive rate (TPR) and the false positive rate (FPR) is plotted due to the imbalance of the dataset. The confusion matrix also gives a sense of the specific number of patients that were correctly classified as dying or not dying, and the ones that were incorrectly classified.

A large area under the curve indicates high recall and precision, with high precision indicating a low false-positive rate and high recall indicating a low false-negative rate. High scores for both indicate that the classifier is producing correct (high precision) results as well as a majority of all positive outcomes (high recall). From Figure 15, it can be seen that there is a large area under the curve, indicating that the model is functioning well. With the best performing model here also achieving an AUC of 96.30%, the model can distinguish between most of the patients who died, and the patients that did not die. It can be seen from the confusion matrix in Fig. 16 that the model correctly predicted 89 individuals who died from the virus and 78 individuals who did not die from the virus. A total of 7 individuals were incorrectly classified as not dying from the virus when they died, and 1 individual was classified as dead when the individual did not die from the virus.

Figure 15 Precision-Recall curve of the best TabNet model for predicting mortality.

Figure 16 Confusion matrix of the best TabNet model for predicting mortality.

The proposed model does an excellent job in predicting mortality. Next, the proposed model will be compared with the baseline models existing in the literature.

It can be seen from Table 14 that the proposed model beats the model reported by Li et al. (2020b) in all metrics, which is a clear suggestion that the proposed model is superior.

Table 14 Comparison of results between proposed method and existing technique.

Model	AUC	F1Score	Accuracy	Recall	Precision	
Li et al. (2020b) (baseline)	84.4	61.6	85.3	70.6	52.2	
Proposed method	96.3	95.8	96.0	99.8	91.8	

Discussion

The main purpose of this study was to develop a deep learning model to predict mortality rate and ICU admission likelihood of patients with COVID-19, and to determine which patient attributes are most important in determining the mortality, and ICU admission of COVID patients. From the results obtained, it can be concluded that:

Finding 1 The proposed TabNet model can predict ICU admission likelihood rate with an AUC of 88.3%, and mortality rate with AUC of 96.3% which beats the model existing in the literature (Li et al., 2020b) by all metrics.

Finding 2 In predicting ICU admission likelihood, the TabNet model depicted that Ferritin, ALT, and Cxdhx were the top 3 predictors of a patient needing ICU admission after contracting COVID 19, and COPD, ferritin, and Myalgia were the top 3 predictors of a patient dying from the COVID-19 disease.

To derive these results, a dataset was chosen. The data in itself was very unbalanced since most individuals who had COVID actually were not rushed to the ICU and certainly did not die. The dataset was pre-processed. The dataset was experimented with 2 sampling techniques which were the ADASYN, and SMOTE technique. Thus, we exclude the oversampling using SMOTE and focus on the balancing using ADASYN.

Table 15 shows the output of the model with varying width of decision prediction layer (nd). The value of nd was changed from a range of 2 to 64 to determine the best output. The results were the best when nd was set to 64.

Table 15 Varying width of decision prediction layer (nd).

nd	AUC	F1Score	Accuracy	Recall	Precision	
Li et al. (2020b) (baseline)	72.8	55.1	72.1	76.0	43.2	
TabNet Baseline (default = 8)	82.9	84.9	83.3	88.7	81.4	
TabNet with nd = 2	76.6	80.8	77.4	89.9	73.39	
TabNet with nd = 4	74.47	80.2	75.6	93.3	70.3	
TabNet with nd = 16	83.27	84.3	83.3	84.3	84.3	
TabNet with nd = 32	83.9	86.6	84.5	94.4	80	
TabNet with nd = 64	84.8	86.5	85.1	89.9	83.3	

Table 16 shows the output of the model with varying number of steps in the architecture (nsteps). The value of nsteps was varied from 3 to 12 to determine the best output. A value of 3 gives the best results. Changing the nsteps to numbers between 8 and 12 showed a slight decrease in performance which indicates that the performance will not be enhanced by increasing the number of steps.

Table 16 Varying number of steps in the architecture (nsteps).

nsteps	AUC	F1Score	Accuracy	Recall	Precision	
Li et al. (2020b) (baseline)	72.8	55.1	72.1	76.0	43.2	
TabNet Baseline (default nsteps = 3)	82.9	84.9	83.3	88.7	81.4	
TabNet with nsteps = 4	76.4	79.1	76.8	83.1	75.5	
TabNet with nsteps = 6	50	69.3	52.9	99.5	52.9	
TabNet with nsteps = 10	80.2	83.8	81	93.3	76.1	

Table 17 shows the output of the model with varying gamma. Changing the gamma shows a very haphazard trend in performance. The best results are given when the gamma is 2.0. Increasing the gamma does not improve the results. Thus, gamma was not increased any further.

Table 17 Varying gamma.

gamma	AUC	F1Score	Accuracy	Recall	Precision	
Li et al. (2020b) (baseline)	72.8	55.1	72.1	76.0	43.2	
TabNet Baseline (default gamma = 1.3)	82.9	84.9	83.3	88.7	81.4	
TabNet with gamma = 1.5	79.5	83.4	80.4	93.3	75.5	
TabNet with gamma = 1.7	77.4	80.6	77.9	86.5	75.5	
TabNet with gamma = 1.9	50	69.3	52.9	100	52.9	

Table 18 shows the output of the model with varying number of independent gates (nindependent). The number of independent gates was varied from 2 to 7. All the nindependent gates obtained similar results converging to the best result with 2. Thus, 2 independent gates give the best results.

Table 18 Varying nindependent.

nindependent	AUC	F1Score	Accuracy	Recall	Precision	
Li et al. (2020b) (baseline)	72.8	55.1	72.1	76.0	43.2	
TabNet Baseline (nindependent = 2)	82.9	84.9	83.3	88.7	81.4	
TabNet with nindependent = 3	50	69.3	52.9	100	52.9	
TabNet with nindependent = 4	82.8	83.4	82.7	82.0	84.9	
TabNet with nindependent = 5	78.5	82.1	79.2	89.9	75.5	

Table 19 gives the output of the model with varying number of shared gates (nshared). The nshared gates was varied from 2 to 7, with 2 gates achieving the best results. Increasing the gates did not improve the results.

Table 19 Varying nshared.

nshared	AUC	F1Score	Accuracy	Recall	Precision	
Li et al. (2020b) (baseline)	72.8	55.1	72.1	76.0	43.2	
TabNet Baseline (nshared = 2)	82.9	84.9	83.3	88.7	81.4	
TabNet with nshared = 3	76.8	80.2	77.4	86.5	74.8	
TabNet with nshared = 4	50	0	47	0	0	
TabNet with nshared = 5	82.8	85.3	83.3	91.0	80.2	

Table 20 presents the output of the model with varying values of momentum. Momentum values of 0.2, and 0.3 displayed the best results, with higher values of momentum producing poorer results.

Table 20 Varying momentum.

momentum	AUC	F1Score	Accuracy	Recall	Precision	
Li et al. (2020b) (baseline)	72.8	55.1	72.1	76.0	43.2	
TabNet Baseline (momentum = 0.02)	82.9	84.9	83.3	88.7	81.3	
TabNet with momentum = 0.1	83.6	85.4	83.9	88.7	82.3	
TabNet with momentum = 0.2	84.6	86.9	85.1	93.3	81.4	
TabNet with momentum = 0.3	84.6	86.8	85.1	93.3	81.4	

Table 21 gives the output of the model with varying lambda sparse. The values of lambda sparse was varied from 0.001 to 0.005. The results of the model showed a negative correlation with the value of lambda sparse. The best result was achieved with a value of 0.001.

Table 21 Varying lambda sparse.

lambda sparse	AUC	F1Score	Accuracy	Recall	Precision	
Li et al. (2020b) (baseline)	72.8	55.1	72.1	76.0	43.2	
TabNet Baseline with lambda sparse = 0.01	82.9	84.9	83.3	88.7	81.4	
TabNet with lambda sparse = 0.01	76.6	80.8	77.4	89.9	73.4	
TabNet with lambda sparse = 0.1	82.2	84.4	82.7	91.0	79.4	

Table 22 gives the output of the model with different types of masks used. It can be concluded from the table that the mask type of entmax gives a better result across the board in all the performance metrics.

Table 22 Varying mask type.

mask type	AUC	F1Score	Accuracy	Recall	Precision	
Li et al. (2020b) (baseline)	72.8	55.1	72.1	76.0	43.2	
TabNet Baseline (mask type = sparsemax)	82.9	84.9	83.3	88.7	81.4	
TabNet with mask type = entmax	87.1	88.9	87.5	94.4	84.0	

Table 23 shows the impact that number of epochs and stopping condition has on the performance of the TabNet architecture. The results are generally better when the stopping condition is defined. The best results are achieved with an epoch of 150, and patience greater than 60. The results do not change when the patience is greater than 60.

Table 23 Impact of number of epochs and stopping condition on the performance of the TabNet architecture.

Model	AUC	F1Score	Accuracy	Recall	Precision	
Li et al. (2020b) (baseline)	72.8	55.1	72.1	76.0	43.2	
TabNet Baseline (epoch = 100)	78.6	81.9	79.2	88.8	76.0	
TabNet Baseline (epoch = 50)	55.2	71.0	57.7	97.8	55.8	
TabNet Baseline (epoch = 150)	82.6	82.4	81.9	84.8	81.3	
TabNet Baseline (epoch = 200)	82.9	81.4	82.9	88.8	81.3	
TabNet Baseline epoch = 150, patience = 5	50	69.3	52.3	100	53.0	
TabNet Baseline epoch = 150, patience = 15	50	69.3	52.3	100	53.0	
TabNet Baseline epoch = 150, patience = 30	82.6	83.4	83.9	88.7	81.3	
TabNet Baseline epoch = 150 patience = 60	83.6	85.4	83.9	88.8	82.3	
TabNet Baseline epoch = 150, patience = 90	83.6	85.4	83.9	88.8	82.3	
TabNet Baseline epoch = 150, patience = 120	83.6	85.4	83.9	88.8	82.3	

Table 24 shows the impact of different dimensionality reduction methods on the performance of the TabNet baseline model. The results using the Fast ICA has the best results with it falling short only on recall to PCA. The most important parameter in the table is the AUC, in which the Fast ICA has a score of 83.6.

Table 24 Comparison of the prediction performance of our baseline models with different variations of dimensionality reduction methods.

Model	AUC	F1Score	Accuracy	Recall	Precision	
Li et al. (2020b) (baseline)	72.8	55.1	72.1	76.0	43.2	
TabNet Baseline + PCA	81.0	83.8	81.5	89.9	78.4	
TabNet Baseline + Fast ICA	83.6	85.4	83.9	88.8	82.3	
TabNet Baseline + Factor Analysis	72.8	75.9	73.2	79.8	72.4	
TabNet Baseline + tSNE	58.2	55.9	57.7	50.6	62.5	
TabNet Baseline + UMAP	54.2	42.3	53.0	32.6	60.4	

The Fast ICA technique yielded the best results, achieving a 2.58% difference over the next highest technique (PCA), in predicting ICU admissions, which can be seen in Table 25.

Table 25 Comparison of the prediction performance of our best TabNet models with different variations of dimensionality reduction methods.

Model	AUC	F1Score	Accuracy	Recall	Precision	
Li et al. (2020b) (baseline)	72.8	55.1	72.1	76.0	76.0	
TabNet Best + PCA	86.2	88.8	86.9	97.8	81.3	
TabNet Best + Fast ICA	86.4	88.5	86.9	95.5	82.5	
TabNet Best + Factor Analysis	82.0	85.3	82.7	94.4	77.8	
TabNet Best + tSNE	60.9	64.5	61.3	66.3	62.8	
TabNet Best + UMAP	58.0	61.5	58.3	62.9	60.2	

Table 26 shows the impact of different oversampling methods on the performance of the TabNet Baseline model. The results using the ADASYN method has the best outcome in all the measured performance metrics.

Table 26 Comparison of the performance of the TabNet Baseline model with oversampling methods.

Model	AUC	F1Score	Accuracy	Recall	Precision	
Li et al. (2020b) (baseline)	72.8	55.1	72.1	76.0	43.2	
TabNet Baseline + SMOTE	79.7	80.7	79.6	85.5	76.3	
TabNet Baseline + ADASYN	83.6	85.4	83.9	88.8	82.3	

The ADASYN gave the best results, achieving a 6.48% difference over the SMOTE technique, which can be seen in Table 27 in predicting ICU admission.

Table 27 Comparison of the performance of the TabNet Best model with oversampling methods.

Model	AUC	F1Score	Accuracy	Recall	Precision	
Li et al. (2020b) (baseline)	72.8	55.1	72.1	76.0	43.2	
TabNet Best + SMOTE	82.1	83.1	82.0	89.2	77.9	
TabNet Best + ADASYN	87.6	89.5	88.1	95.5	84.2	

Table 28 shows the output of the model with varying width of decision prediction layer (nd). The value of nd was changed from a range of 2 to 64 to determine the best output. The results were the best when nd was set to 64.

Table 28 Varying width of decision prediction layer (nd).

Model	AUC	F1Score	Accuracy	Recall	Precision	
Li et al. (2020b) (baseline)	84.4	61.6	85.3	70.6	52.2	
TabNet Baseline (default = 8)	90.4	89.5	89.7	97.5	82.3	
TabNet with nd = 2	81.2	78.9	81.7	75.9	82.2	
TabNet with nd = 4	83.3	81.8	83.4	82.3	81.3	
TabNet with nd = 16	90.3	89.4	89.7	96.2	83.5	
TabNet with nd = 32	83.9	86.6	84.5	94.4	80	
TabNet with nd = 64	84.8	86.5	85.1	89.9	83.3	

Table 29 shows the output of the model with varying number of steps in the architecture (nsteps). The value of nsteps was varied from 3 to 12 to determine the best output. A value of 3 gives the best results. Changing the nsteps to numbers between 8 and 12 showed a slight decrease in performance which indicates that the performance will not be enhanced by increasing the number of steps.

Table 29 Varying the number of steps in the architecture (nsteps).

Model	AUC	F1Score	Accuracy	Recall	Precision	
Li et al. (2020b) (baseline)	84.4	61.6	85.3	70.6	52.2	
TabNet Baseline (default nsteps = 3)	90.4	89.5	89.7	97.5	82.3	
TabNet with nsteps = 4	85.9	84.7	85.7	87.3	82.1	
TabNet with nsteps = 6	87.8	86.6	88.0	86.1	87.2	
TabNet with nsteps = 10	73.0	70.8	73.1	72.2	69.5	

Table 30 gives the output of the model with varying values of gamma. The values of gamma was varied from 1.3 to 2.2. The performance of the model shows a very haphazard trend with the changing values. A gamma of 2.0 gives the best results, and increasing the gamma any further did not improve the results.

Table 30 Varying gamma.

Model	AUC	F1Score	Accuracy	Recall	Precision	
Li et al. (2020b) (baseline)	84.4	61.6	85.3	70.6	52.2	
TabNet Baseline (default gamma = 1.3)	90.4	89.5	89.7	97.5	82.3	
TabNet with gamma = 1.5	87.7	86.5	88.0	84.8	88.2	
TabNet with gamma = 1.7	92.6	91.8	92.0	98.7	85.7	
TabNet with gamma = 1.9	90.0	89.2	89.7	93.7	85.1	

Table 31 gives the output of the model with varying number of independent gates (nindependent). The number of independent gates was varied from 2 to 7. The number of gates which gave the best output is 2, and increasing the number of gates decreased the accuracy of the results.

Table 31 Varying the number of independent gates (nindependent).

Model	AUC	F1Score	Accuracy	Recall	Precision	
Li et al. (2020b) (baseline)	84.4	61.6	85.3	70.6	52.2	
TabNet Baseline (nindependent = 2)	90.4	89.5	89.7	97.5	82.3	
TabNet with nindependent = 3	86.5	85.7	85.7	94.9	78.2	
TabNet with nindependent = 4	82.8	83.4	82.7	82.0	84.9	
TabNet with nindependent = 5	85.6	84.3	85.7	84.8	83.8	

Table 32 gives the output of the model with varying number of shared gates (nshared). The number of shared gates was varied from 2 to 7. The number of shared gates of 2 gave the best results and increasing the number of gates did not improve the results. Although there is a spike in results when the number of shared gates is 5, the performance reduces when it is increased further.

Table 32 Varying the number of shared gates (nshared).

Model	AUC	F1Score	Accuracy	Recall	Precision	
Li et al. (2020b) (baseline)	84.4	61.6	85.3	70.6	52.2	
TabNet Baseline (nshared = 2)	90.4	89.5	89.7	97.5	82.3	
TabNet with nshared = 3	88.9	87.7	89.1	86.1	89.5	
TabNet with nshared = 4	87.5	86.4	87.4	88.6	84.3	
TabNet with nshared = 5	90.2	89.3	90.3	89.9	88.8	

Table 33 gives the output of the model with varying vales of momentum. The value of momentum was varied from 0.02 to 0.3, with 0.02 giving the best results. Increasing the value of the momentum gave poorer results.

Table 33 Varying momentum.

Model	AUC	F1Score	Accuracy	Recall	Precision	
Li et al. (2020b) (baseline)	84.4	61.6	85.3	70.6	52.2	
T Li, Xiaoran et al. (baseline) Li et al. (2020b)	84.4	61.6	85.3	70.6	52.2	
TabNet Baseline (momentum = 0.02)	90.4	89.5	89.7	97.5	82.3	
TabNet with momentum = 0.1	89.9	89.0	89.1	97.5	81.9	
TabNet with momentum = 0.2	87.1	86.2	86.3	94.5	78.9	
TabNet with momentum = 0.3	88.8	86.0	88.0	97.5	80.2	

Table 34 gives the output of the model with varying values of lambda sparse. The values of lambda sparse was varied from 0.001 to 0.005, with 0.001 achieving the best results. The value of the lambda sparse had a negative correlation with the performance of the model.

Table 34 Varying lambda sparse.

Model	AUC	F1Score	Accuracy	Recall	Precision	
Li et al. (2020b) (baseline)	84.4	61.6	85.3	70.6	52.2	
TabNet Baseline (lambda sparse = 1e−3)	90.4	89.5	89.7	97.5	82.3	
TabNet with lambda sparse = 1e−2	88.5	87.6	88.0	93.7	82.2	
TabNet with lambda sparse = 1e−1	88.6	87.5	88.6	88.6	86.4	

Table 35 gives the output of the model with different mask types. The different masktypes used were sparsemax, and entmax. The output using the sparsemax had a better result compared to the entmax.

Table 35 Varying mask type.

Model	AUC	F1Score	Accuracy	Recall	Precision	
Li et al. (2020b) (baseline)	84.4	61.6	85.3	70.6	52.2	
TabNet Baseline (masktype = sparsemax)	90.4	89.5	89.7	97.5	82.3	
TabNet with mask type = entmax	87.0	86.3	85.7	99.8	76.0	

Table 36 shows the impact that number of epochs and stopping condition has on the performance of the TabNet architecture. The results are always better when a stopping condition is defined. The best results are achieved with an epoch of 150, and patience greater than 60. The results do not change when the patience is greater than 60.

Table 36 Impact of number of epochs and stopping condition on the performance of the TabNet architecture.

Model	AUC	F1Score	Accuracy	Recall	Precision	
Li et al. (2020b) (baseline)	84.4	61.6	85.3	70.6	52.2	
TabNet Baseline (epoch = 100)	90.4	89.5	89.7	97.5	82.3	
TabNet Baseline (epoch = 50)	85.9	85.1	85.1	93.7	77.9	
TabNet Baseline (epoch = 150)	90.4	89.5	89.7	97.5	82.3	
TabNet Baseline (epoch = 200)	90.4	89.5	89.7	97.5	82.3	
TabNet Baseline maximum epoch = 150 with early stopping, patience = 5	85.6	84.5	85.1	89.9	78.0	
TabNet Baseline maximum epoch = 150 with early stopping, patience = 15	85.5	84.7	84.6	94.5	76.5	
TabNet Baseline epoch = 150 with early stopping, patience = 30	90.4	89.5	89.7	97.5	82.3	
TabNet Baseline epoch = 150 with early stopping, patience = 60	91.2	90.2	90.1	97.6	84.1	
TabNet Baseline epoch = 150 with early stopping, patience = 90	91.2	90.2	90.1	97.6	84.1	
TabNet Baseline epoch = 150 with early stopping, patience = 120	91.2	90.2	90.1	97.6	84.1	

Table 37 shows the impact of different dimensionality reduction methods on the performance of the TabNet baseline model. The results using the PCA is the best with it falling short only on recall to Fast ICA. The most important parameter in the table is the AUC, in which the PCA has a score of 94.0.

Table 37 Comparison of the prediction performance of our baseline models with different variations of dimensionality reduction methods.

Model	AUC	F1Score	Accuracy	Recall	Precision	
Li et al. (2020b) (baseline)	84.4	61.6	85.3	70.6	52.2	
TabNet Baseline + PCA	94.0	93.3	93.7	97.5	89.5	
TabNet Baseline + Fast ICA	93.8	92.9	93.1	99.7	86.8	
TabNet Baseline + Factor Analysis	90.4	89.5	89.7	97.5	82.3	
TabNet Baseline + tSNE	72.7	73.2	71.4	86.1	63.6	
TabNet Baseline + UMAP	69.2	68.6	68.6	75.9	62.5	

Table 38 shows the impact of different dimensionality reduction method on the performance of the best TabNet model. The results are better across all the measured performance metrics when the Fast ICA is used.

Table 38 Comparison of the prediction performance of our best TabNet models with different variations of dimensionality reduction methods.

Model	AUC	F1Score	Accuracy	Recall	Precision	
Li et al. (2020b) (baseline)	84.4	61.6	85.3	70.6	52.2	
TabNet Best + PCA	94.2	93.4	93.7	98.7	88.6	
TabNet Best + Fast ICA	95.3	94.6	94.9	99.8	89.8	
TabNet Best + Factor Analysis	93.1	92.3	92.3	98.7	86.7	
TabNet Best + tSNE	73.7	73.6	72.6	84.8	65.0	
TabNet Best + UMAP	66.7	69.0	64.6	88.6	56.9	

Table 39 shows the impact of different oversampling methods on the performance of the TabNet Baseline model. The results using the ADASYN method has the best results in all the metrics except for precision where the SMOTE method is better.

Table 39 Comparison of the performance of the TabNet Baseline model with oversampling methods.

Model	AUC	F1Score	Accuracy	Recall	Precision	
Li et al. (2020b) (baseline)	84.4	61.6	85.3	70.6	52.2	
TabNet Baseline + SMOTE	93.2	92.3	92.1	97.2	88.7	
TabNet Baseline + ADASYN	93.8	92.9	93.1	99.3	86.8	

Table 40 shows the impact of different oversampling methods on the performance of the TabNet Best model. The results using the ADASYN method has the best results in all the measured performance metric.

Table 40 Comparison of the performance of the TabNet Best model with oversampling methods.

Model	AUC	F1Score	Accuracy	Recall	Precision	
Li et al. (2020b) (baseline)	84.4	61.6	85.3	70.6	52.2	
TabNet Best + SMOTE	94.3	94.6	94.3	98.8	90.6	
TabNet Best + ADASYN	96.3	95.8	96.0	99.8	91.8	

Five different dimensionality techniques were also experimented with to improve the results. The other dimensionality reduction techniques are excluded, and the Fast ICA technique is concentrated upon to achieve the final results.

The various hyperparameters were also tuned, and the best results of each was combined with the dimensionality technique and then oversampled to obtain the final result for all metrics. Results achieved were, an AUC of 88.3%, F1 score of 89.7%, the accuracy of 88.7%, recall of 93.3%, and precision of 86.4% for predicting ICU. In predicting mortality, results of 96.3% AUC, 95.8% F1 score, accuracy of 96.0%, recall of 99.8%, and precision of 91.8% were obtained. The reason why the results in predicting mortality achieves higher performances than the one in predicting ICU admission could be because sometimes individuals that need ICU admission, do not get the opportunity due to lack of beds available at that time because of large volumes of individuals present at the hospital needing the same resources. In the case of mortality, when an individual dies, the individual dies, there is no middle ground, so it is relatively easier to distinguish mortality than ICU admission.

A confusion matrix was constructed to show specifics, where there were more false positives than false negatives in both determining ICU admission and mortality. The reason for more false positives than false negatives could be because doctors have to make a quick and instant guess as to which patient needs the ICU at that time by simply looking at the physical conditions of the patient present. Due to the lack of time and resources, they depend on only those physical symptoms to make a decision, so patients who can deteriorate quickly due to underlying illnesses or other factors are often overlooked for admission to the ICU simply because they do not show physical deterioration at the time of decision making.

The process by which the proposed model makes decisions to determine which features are most important was also determined. The model uses Masks which shows the features they were paying the most attention to in the heat map, which can be seen here in Figs. 4 and 11. This was then used to construct the global feature importance graph, which is easier to understand, where the longer the bar, the more importance it has in determining if a patient with COVID-19 is likely to be sent to the ICU or if the patient is likely to die from the disease.

The findings from our model suggesting the most important features in predicting ICU admission and mortality, has been supported by other literature’s, these are shown below.

Ferritin is the symptom of the patient which is the most important in determining if the patient needs ICU or not. Ferritin represents how much iron is contained in the body, and if a ferritin test reveals a lower-than-normal ferritin level in the blood, this may indicate that the body’s iron stores are low. This is a high indication of iron deficiency which can cause anaemia (Dinevari et al., 2021). Ferritin levels were found to be elevated upon hospital admission and throughout the hospital stay in patients admitted to the ICU by COVID-19. In comparison to individuals with less severe COVID-19, ferritin levels in the peripheral blood of patients with severe COVID-19 were shown to be higher. As a result, serum ferritin levels were found to be closely linked to the severity of COVID19 (Dahan et al., 2020). Early analysis of ferritin levels in patients with COVID-19 might effectively predict the disease severity (Bozkurt et al., 2021). The magnitude of inflammation present at admission of COVID-19 patients, represented by high ferritin levels, is predictive of in-hospital mortality (Lino et al., 2021). Studies indicate that Chronic obstructive pulmonary disease (COPD) is the symptom that shows the most importance in predicting mortality among COVID-19 patients. This COPD is a chronic inflammatory lung condition in which the lungs’ airflow is impeded. Breathing difficulties, cough, mucus (sputum) production, and wheezing are all symptoms. Since COVID-19 is a disease that affects the respiratory system, it makes sense that a disease like COPD which also affects the lungs could have devasting effects on a patient who contracts COVID-19 (Gerayeli et al., 2021). Patients with Chronic Obstructive Pulmonary Disease (COPD) have a higher prevalence of coronary ischemia and other factors that put them at risk for COVID-19-related complications. The results of this study confirm a higher incidence of COVID-19 in COPD patients and higher rates of hospital admissions (Graziani et al., 2020). While COPD was present in only a few percentage of patients, it was associated with higher rates of mortality (Venkata & Kiernan, 2020). It can be observed that the top 3 highest predictors for mortality and ICU admission are different, which indicates that there are some features which can be seen in the later and more advanced stages of COVID (at the time of death). For example, Pardhan et al. (2021) observed that COPD is reported more often than asthma, suggesting that physicians in Sweden considered COPD to be a better predictor than asthma for detecting severe COVID-19 cases.

It can also be seen that shortness of breath had a high correlation with ICU admission but it was not among the top predictors for predicting ICU admission. This is because correlation looks at only the linear relationship between that feature and the target without considering other features. For example, a single feature can have a low correlation, but when combined with other features, it can offer a high predictive power, as in the case of the COPD.

The limitations of this study are that the sample size is small, with only about 1,000 patients included in the study. The study was restricted to patients at Stony Brook University Hospital and conducted between 7 February, 2020 to 4 May, 2020.

Conclusion

This paper proposes a tabular, interpretable deep learning model to predict ICU admission likelihood and mortality of COVID-19 patients. The proposed model achieves this by employing a sequential attention mechanism that selects the features at each step of the decision-making process based on a sparse selection of the most important features such as patient demographics, vital signs, comorbidities, and laboratory discoveries.

ADASYN was used to balance the data sets, Fast ICA to extract useful features, and all the various hyperparameters tuned to improve results. The proposed model achieves an AUC of 88.3% for predicting ICU admission likelihood which beats the 72.8% reported in the literature. The proposed model also achieves an AUC of 96.3% for predicting mortality rate which beats the 84.4% reported in the literature. The most important patient attributes for predicting ICU admission and mortality were also determined to give a clear indication of which attributes contribute the most to a patient needing ICU and a patient dying from COVID-19, where these claims were also backed up previous studies as well. The information from the model can be used to assist medical personnel globally by helping direct the limited healthcare resources in the right direction, in prioritizing patients, and to provide tools for front-line doctors to help classify patients in time-bound and resource-limited scenarios.

For future work, the study can be extended to include a lot more patients over a longer time frame from several hospitals. The proposed method can be combined with other machine learning methods for improved results. This study could be extended to include more diseases, allowing the healthcare system to respond more quickly in the event of an outbreak or pandemic.

Supplemental Information

Supplemental Information 1 The source code for producing Tables 29 to 42 and Figures 11 to 16, in which the hyperparameter values have been manually tuned for each experiment.

Click here for additional data file.

Supplemental Information 2 The source code for producing Figure 3.

Click here for additional data file.

Supplemental Information 3 The source code for producing Figure 10.

Click here for additional data file.

Supplemental Information 4 The source code for producing Tables 5 to 10.

Click here for additional data file.

Supplemental Information 5 The source code for producing Tables 14 to 28, and Figures 4 to 9, in which the hyperparameter values have been manually tuned for each experiment.

Click here for additional data file.

Additional Information and Declarations

Competing Interests

Author Contributions

Data Availability

The authors declare that they have no competing interests.

Amril Nazir conceived and designed the experiments, performed the experiments, analyzed the data, performed the computation work, prepared figures and/or tables, authored or reviewed drafts of the paper, and approved the final draft.

Hyacinth Kwadwo Ampadu conceived and designed the experiments, performed the experiments, analyzed the data, performed the computation work, prepared figures and/or tables, authored or reviewed drafts of the paper, and approved the final draft.

The following information was supplied regarding data availability:

The study comprised medical information of patients from the Stony Brook University Hospital such as the demographics, comorbidities, symptoms, vital signs, and laboratory tests recorded, which were conducted between 7 February, 2020 to 4 May, 2020. The data are available at:

- https://doi.org/10.7717/peerj.10337/supp-1

- https://doi.org/10.7717/peerj.10337/supp-2.

The source code for the figures and tables generated in the article are available in the Supplemental Files.

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
