# Peer review of "Interpretable deep learning for the prediction of ICU admission likelihood and mortality of COVID-19 patients"

_PeerJ Computer Science, doi:10.7717/peerj-cs.889_

## Round 0.1 · original submission · Major Revisions

Reviewer 2 has highlighted flaws in the manuscript which must be addressed.

[# Reviewer 1 ·

Basic reporting

The authors should decrease the number of tables as possible. Some tables could be merged or moved to supplement files, such as hyperparameters tuning. They should keep the most important parts for readers.

Experimental design

The research question is "interpretable". The authors only pick the important features from the model but not to the point "interpretable". They can explain the definition of "interpretable".

Validity of the findings

Nil.

Additional comments

The analytic process could be concise and to the point.

Reviewer 2 ·

Basic reporting

* The report is too detailed and includes trivial information which renders the reading experience to be exhausting. In scientific papers many different methods are used these days and it is necessary to think about what information should be included, what should be cited and what should be left out (as trivial information). This is a process which the current report can greatly benefit from. More focus needs to be directed towards the main findings of the papers and the methods that ended up being used in the most optimal model that was eventually applied.

* The figures and the tables lack detailed descriptions. Furthermore they are not completely labelled ans for instance the x axis and y axis labels are missing in the feature importance plots (i.e. figure 2 and 4).

* information provided in many tables are redundant, as for instance in many tables such as table 6 the "Without symptoms" percentages are the complements of the "With symptoms". Better and more brief means of describing the 'Description of data sets' and the 'statistical analysis' must be used that listing all these information in a sequence of tables. If some of these information are not relevant or significant they can be referred to in supplement documents rather than in the main paper.

* The following sentences require rewriting :

- Line 145: "ADASYN (He et al., 2008) is a synthetic data generation algorithm that has the benefits of not copying
the same minority data and producing more data which are harder to learn."

- In equation 2: Xk is not defined . What is Xk ?

- Line 153: "… same proportion of observations, and also due to the dataset 154 being imbalanced. The fold used was k=5 which means …"

-Top of page 4: "These operations are done sequentially in the order as shown in Equations 4, and 5:
FC =W(x) (10) …"
Do the equations need to be repeated ?! Please clarify how do they relate to formula 4 and 5.

In tables 5 and 8: Both columns are labelled as "no-ICU" !

Line 273: "Overall, over 50% of the individuals acquired a symptom of a disease before they died."
This does not seem correct! How far over 50% ? and by "a symptom" do the authors mean "at least one symptom" ? This sentence suggests that a bit less than 50% of patients died without showing any symptoms ?!

Line 392: "The key performance metric in this analysis is the AUC score
and it can be plotted using the precision-recall curve which demonstrates a trade-off between the recall score (True Positive Rate), and the precision score (Positive Predictive Value)."
AUC is usually referred to the area under ROC curve. Do you mean the area under Precision Recall curve here ? If yes, you can use AUCPR
to avoid confusion.

Experimental design

* I was not convinced and completely informed about how the authors avoided overfitting to the data throughout the whole study. They mention that they consider a 5 fold cross-validation using stratified K-fold :
"The fold used was k=5 which means that the dataset was divided into 5 folds with each fold being utilized once as a testing set, with the remaining k - 1 folds becoming the training set."

However, further down in the report (line 299) they mention : "The ENTIRE dataset is split into training and validation in a 90%-10% ratio respectively, to enable a good number of data points to be trained to get better results. "
If the entire data is divided to train and validation, how is the 5 fold cross-validation implemented ?


* The pvalues and FDRs of the correlations are missing throughout the report. Furthermore, the correlation (distance) method hat was used should be clarified; was it "Pearson" ?

Validity of the findings

* My biggest issue with the study is that as the authors have also mentioned the data is small for the kind of prediction model that the authors are aiming for. Furthermore the data is imbalanced, whilst computational methods have been used to overcome the effects that the imbalance of data may have on the training and testing of the model. All these, together with the fact that only one data was used for the each prediction model (one for ICU prediction of COVID-19 and another for the mortality) makes any findings related to the most effective prediction factors to be imprecise and unreliable if they are solely dependent on the computational methods used by the authors. At least these findings should be directly supported by other litterateurs and studies. As for instance is the direct association of Ferritin with ICU visits or mortality of COVID-19 patients (rather than the indirect association through COPD that is mentioned by the authors) been discovered by other studies as well ?

Additional comments

My criticism regarding the study aside the inclusion and possibly publication of all the codes used for analyzing the data and producing the figures by the authors is an excellent practice that should become more common in the scientific community.

---

## Round 0.2 · Minor Revisions

Reviewer 2 still has concerns, which must be addressed.

Reviewer 1 ·

Basic reporting

no comment

Experimental design

no comment

Validity of the findings

no comment

Additional comments

The authors answered the questions

Reviewer 2 ·

Basic reporting

* This is a suggestion (and it is OPTIONAL!): Figures have titles only and lack descriptions! legends are described in the PeerJ “Instruction for Authors” to be optional, however I think that it would improve the readability of a paper. Legends and table/figure descriptions allow the person who reads it to fully understand the plot/illustration without needing to read the text. As for instance in figure 2 you can mention what the heat map mean (e.g. shows whether a feature is selected at a given decision step in the model), what the x and y are; and what the brighter colours stand for and etc.

Experimental design

* It would be interesting if the authors can elaborate that how can factors such as COPD and Myalgia mortality be amongst 3 highest predictors of mortality but not amongst the highest factors for ICU admission. Are they seen in the later and more advanced stages of Covid?

* I understand that Neural Networks does not necessarily model in the same manner as linear regression and etc, however it would still be inteersting if the authors could elaborate how the Shortness of Breath (SOB) feature which has the highest correlation with admission to the ICU unit is not amongst its highest 3 predictors !

Validity of the findings

* Something that is still not fully clear to me is that how has the stratified k-fold split of 5 been implemented in the training and testing of the models, using the most optimal model for ICU admission as for instance. The applied cross validation (K-fold split of 5) is still with 5 replications; therefore, I assume that 5 sets of training and testing of each prediction model been executed? An ROC curve can be plotted for each prediction on each test subset, so this means that actually for each prediction model 5 PRC/ROC curves are resulted and 5 AUPRC/AUCs are measured? The authors however describe only one PRC/ROC curve and AUPRC/AUC. Are these the PRC/ROC curves of the most optimal run (with the highest AUPRC/AUC)? If that is true, it should be written clearly and optionally the authors may want to mention the variance of the AUPRC/AUC across the 5 cross validation runs, or report the mean AUPRC/AUC as well.

Additional comments

* Correct m_l and m_s in the line beneath equation 1 , beneath line 153 (some lines don’t have numbers !).
* Line 167: “TabNet deep learning model 15 is”. Is “15” a typo ?

---

## Round 0.3 · accepted · Accept

The authors have addressed the reviewer's comments.

Reviewer 2 ·

Basic reporting

No comment.

Experimental design

No comment.

Validity of the findings

No comment.